# Socioeconomic status and subjective well-being: The mediating role of class identity and social activities

**Baoqin Wang**[1,2☯], **Hang Zhao**[1,2☯]**, Hao Shen**[1,2]**, Yi Jiang**[1,2]*

**1** School of Public Health, Chongqing Medical University, Chongqing, China, **2** Research Center for Medical and Social Development, School of Public Health, Chongqing Medical University, Chongqing, China

☯ These authors contributed equally to this work.

* jiangyilaoshi@163.com

## Abstract

### Background

Subjective well-being has a significant impact on an individual's physical and mental health. Socioeconomic status, class identity, and social activity participation play important roles in subjective well-being. Therefore, the aim of this study was to uncover the mechanisms through which these factors influence subjective well-being.

### Methods

A total of 1926 valid samples were recruited using the Chinese General Social Survey 2021 (CGSS 2021). The Chinese Citizen's Subjective Well-Being Scale (SWBS-CC) was employed to assess subjective well-being. Socioeconomic status was measured using income and education, and class identity and social activity participation were measured using Likert scales. Pearson correlation analysis and the chain mediation model were conducted to explore the relationship between these factors. Finally, the Bootstrap method was used to examine the path coefficients.

### Results

A significant correlation was found between socioeconomic status, class identity, social activity, and subjective well-being (p < 0.01). The indirect effect of socioeconomic status on subjective well-being mediated by class identity was 0.351 (95% CI: 0.721, 1.587), while the indirect effect of socioeconomic status on subjective well-being mediated by social activity was 0.380 (95% CI: 0.059, 0.240). The effect mediated by both class status and social activities was 0.011 (95% CI: 0.010, 0.093).

### Conclusions

The study showed that socioeconomic status, class identity, and social activity had significant effects on subjective well-being. Class identity and social activity partially mediated the effects of socioeconomic status on subjective well-being, and they had a chain mediating

**Data Availability Statement:** All data files are available from the Chinese General Social Survey 2021 database(URL:http://www.cnsda.org/index.php?r=projects/view&id=65635422).

**Funding:** The author(s) received no specific funding for this work.

**Competing interests:** The authors have declared that no competing interests exist.

effect between socioeconomic status and subjective well-being. Therefore, policymakers have the opportunity to enhance subjective well-being in lower socioeconomic status groups by promoting individual class identity and encouraging greater social activity participation.

## Introduction

Subjective well-being (SWB) and its determinants have garnered extensive scholarly attention, and research in this area is expanding. The Organization for Economic Co-operation and Development (OECD) and the European Union (EU) advocate for prioritizing the well-being of individuals in policy design [1]. Subjective well-being reflects an individual's social functioning and adaptation to the environment, comprising life satisfaction, the experience of positive emotions, and the absence of negative emotions [2]. Subjective well-being served as a robust predictor of overall health and well-being. Research has shown that SWB has a protective effect on health [3], and individuals with higher SWB tend to experience better health outcomes and longer life expectancy [4]. Furthermore, SWB was an important indicator of successful aging in a country or region [5]. Therefore, it is necessary to extensively explore the factors and potential pathways influencing SWB.

Socioeconomic status (SES) is an essential influencing factor for SWB. Research has shown that SES could contribute to SWB through various external and internal mechanisms. Stress theory could elaborate on the mechanisms and pathways through which SES can influence well-being, especially the concepts of coping resources and stressors. Individuals with higher SES tended to enjoy better life circumstances and more social resources, buffering the effects of adverse events [6, 7]. Furthermore, individuals with higher SES were exposed less to stressful and uncontrollable life events, which were associated with higher SWB [8]. Subjective class identity is an individual's perception of their position in the social class structure, reflecting the individual's internalization, adaptation, and acceptance of the social environment and group [9]. Class identity depends heavily on the degree of individual demand fulfillment. Higher demand fulfillment was associated with higher SWB [10]. Social comparison theory suggests that people determine their values and the realization of their demands by comparing themselves to others [11]. Therefore, the SWB of individuals depends not only on their SES and class identity but also on their perceptions of their position relative to others. Similarly, engaging in social activities played an important role in the well-being of the population. Social support theory emphasizes the importance of social relationships and the support individuals receive from their social networks [12]. Participation in social activities was associated with more active social networks [13, 14]. These social networks provided emotional, informational, and instrumental support that facilitated the reduction of isolation and stress and increased resilience and SWB [15].

Abundant evidence has demonstrated a positive correlation between SES and social class identity. Bourdieu's theory of cultural capital emphasizes the importance of cultural capital in shaping social class identity [16, 17], such as education, cultural resources, and social skills. However, cultural capital acquisition is often influenced by SES [18]. For example, individuals from higher SES backgrounds were more likely to have access to high-quality education, exposure to arts and cultural experiences, and opportunities to develop cultural tastes and skills [19, 20]. The social identity theory also explained the association between SES and class identity. Social identity theory suggests that people tend to categorize themselves based on various social dimensions, especially SES [21]. Individuals with similar SES often have shared

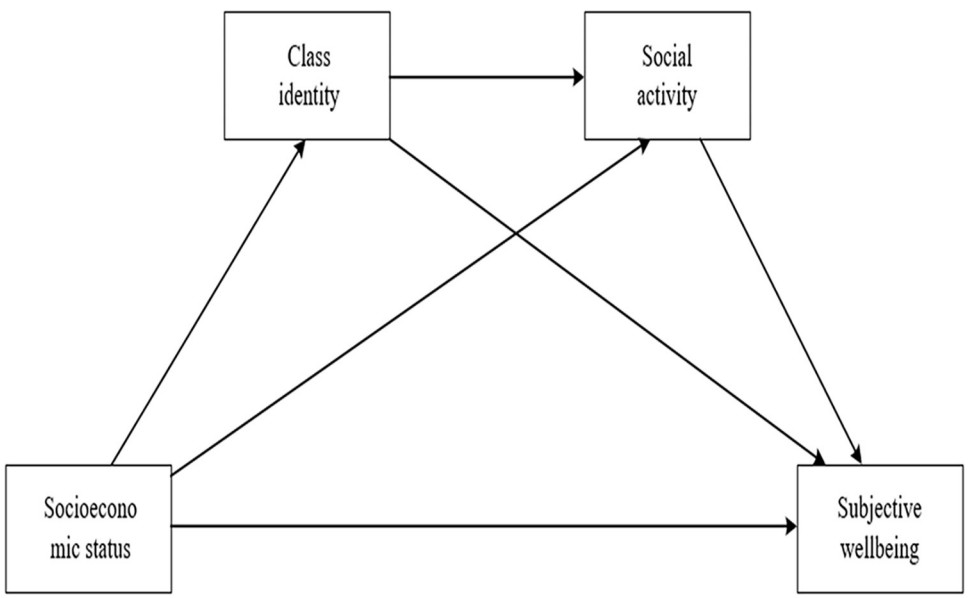

**Fig 1. The conceptual model based on previous research and theory.**

experiences and challenges, contributing to the development of a common class identity. Research has shown the significant and positive impact of SES and class identity on individuals' engagement in social activities [22]. Based on stress theory, higher SES and class identity were associated with more social support and fewer stressors, possibly promoting individuals to engage more in social activities [23]. Individuals with higher SES and class identity typically had access to a wider range of social support and resources, which could facilitate participation in social activities [24, 25].

In conclusion, although the effect of SES on SWB has been explored, few previous scholars have examined the mediating role of class identity and social activities. Therefore, we used the chain mediation model to examine the potential relationship among SES, class identity, social activity, and SWB. The above theories and literature have revealed the interconnections among SES, class identity, and social activities, emphasizing their importance as influences on SWB. Based on these, the conceptual framework (shown in Fig 1) and the following hypotheses were proposed: 1) there is a significant association between SES, class identity, participation in social activities, and SWB; 2) class identity and participation in social activities mediate between SES and SWB; 3) class identity and social activity participation have a chain-mediated role between SES and SWB.

## Methods

### Data

The data available for this study was from the Chinese General Social Survey of 2021 (CGSS 2021), a nationwide program that commenced in 2003 to investigate various social, political, economic, and cultural changes in Chinese society. This comprehensive and continuous database was collected systematically at multiple levels, ranging from Chinese individuals and families to communities and social groups. In 2021, the CGSS adopted a multi-stage stratified probability sampling method and covered 30 provinces in mainland China to ensure good scientific and national representativeness.

The CGSS was conducted in accordance with the ethical principles of the Declaration of Helsinki. The Ethics Committees of Renmin University of China and Hong Kong University of Science and Technology are responsible for ethical approval and consent to participate. We received authorization to use the publicly accessible CGSS. This study was conducted based on de-identified publicly available CGSS data, which is available at http://cgss.ruc.edu.cn/. Therefore, ethical approval or informed consent was no longer required for this study.

The total number of survey samples was 8,148. The CGSS 2021 collected data on demographic characteristics, SES, class identity, social activity, and SWB, which provided the basis for our study. After excluding observations with important missing data, a total of 1926 participants were included in this study (18 years and above).

## Measure

Subjective Well-being was measured using the Subjective Well-Being Scale for Chinese Citizens (SWBS-CC), which was proposed by mainland Chinese scholar Xing Zhanjun [26]. The scale includes 20 items covering dimensions such as mental health, social confidence, growth and progress, goal value, self-acceptance, physical health, mental balance, and interpersonal adjustment, which is a valid measure for studies of urban residents in mainland China [27, 28]. The scale exhibited strong reliability (Cronbach's alpha = 0.838) and validity (KMO = 0.864). Participants rated the scale using a 6-point Likert scale, and the total score was computed by summing all the responses. The range of SWB was 20–120, with higher total scores indicating better well-being.

The present study employed SES as a key variable. Following relevant studies, we employed education and income as measures of SES [29–31]. Education was categorized as follows: (1) uneducated, (2) primary school and below, (3) middle school, (4) high school, and (5) college and higher. Personal income was stratified into quantities (1–5) based on annual earnings [32]. To create the SES composite score, standardized z-scores for education and income were averaged.

According to social identity theory, social identity derives primarily from group membership or qualifications and is rooted in people's subjective perceptions of their class identity. To measure class identity, self-rated class identity was used based on previous research [33]. The CGSS asked the question about class identity: "In general, where do you personally stand in the current society?" Responses were recorded on a 10-point Likert scale, with higher scores indicating higher levels of class identity.

Based on previous studies [34, 35], the social activity of participants in the past year was measured by 12 items with acceptable reliability (Cronbach's alpha = 0.674) and validity (KMO = 0.782). The questions focused on outdoor leisure activities, including watching television or videos, going to the cinema, shopping, reading books/newspapers/magazines, attending cultural events, gathering with relatives, meeting with friends, listening to music at home, participating in physical activity, watching live sports, doing crafts, and surfing the Internet. These entries were rated using a 5-point Likert scale. Answers were set to (1) never, (2) several times a year or less, (3) several times a month, (4) several times a week, and (5) every day. The total score of social activities was calculated by summing the 12 entries. The range of social activities was 12–60, with higher total scores implying higher levels of activity.

We included several control variables to improve the estimation accuracy, including age, gender, ethnicity, marriage, household registration, and self-rated health. Ethnicity was divided into Han and others, and household registration was divided into urban and rural. Marital status was classified as: unmarried, divorced, and widowed; married and having a

spouse. The self-assessment of health was based on the question, "How do you feel about your current state of health?", with values from 1 (very poor) to 5 (very good).

## Statistical analysis

All statistical analyses were conducted in STATA version 17.0. A P-value of less than 0.05 was considered statistically significant. Statistical descriptions were described by the mean and standard deviation (for continuous variables) and frequency and percentages (for categorical variables). Pearson correlation analysis was used to investigate the correlation between key variables. A chain mediation model analysis was employed to explore the relationship between SES, class identity, social activity, and SWB. Control variables were included in all path analyses to improve the reliability of the estimation results. Considering the potential heteroskedasticity problem in the cross-sectional data, we used robust standard errors for estimation. Finally, the total and indirect effects and their 95% confidence intervals (CI) were tested using bootstrapping with 1000 iterations.

## Results

### Descriptive statistics

Table 1 displays the descriptive statistics of the participants. A total of 1926 individuals were selected from the Chinese General Social Survey 2021 (CGSS 2021). Of these, 883 (45.85%) were male and 1043 (54.15%) were female. More than 90% of the participants were of Han Chinese ethnicity. Additionally, approximately 75% were either married or had a spouse, and

**Table 1. Descriptive statistics of participant characteristics.**

| Variable | Mean (SD)/ Frequency (%) |
|---|---|
| Socioeconomic status | 0.00 (0.84) |
| Class identity | 4.35(1.86) |
| Social activities | 28.22(6.58) |
| Subjective well-being | 86.99(11.73) |
| Age (years) | 49.93(16.80) |
| Gender | |
| Female | 1043 (54.15) |
| Male | 883(45.85) |
| Ethnic | |
| Non-Han | 130(6.75) |
| Han | 1796 (93.25) |
| Marriage | |
| Unmarried | 479(24.87) |
| Married | 1447(75.13) |
| Household registration | |
| Rural | 1310(68.02) |
| Urban | 616(31.98) |
| Self-rated health | |
| Very poor | 85(4.41) |
| Poor | 236(12.25) |
| Fair | 543(28.19) |
| Good | 696(36.14) |
| Very good | 366(19.00) |

approximately 68% resided in rural areas. The percentages of participants with very poor, poor, fair, good, and very good health were 4.41%, 12.25%, 28.19%, 36.14%, and 19.00%, respectively. The mean of age, SES, class identity, social activity, and SWB were 49.43, 0.00, 4.35, 28.22, and 86.99, respectively.

## Correlation analysis

As shown in Table 2, the results of the correlation analysis indicated the mutually significant correlations between SES, class identity, social activity, and SWB (p < 0.001). Consistent with our hypotheses, SES showed positive correlations with class identity (r = 0.130, p < 0.001), social activity (r = 0.486, p < 0.001), and SWB (r = 0.180, p < 0.001). Class identity was positively associated with social activity (r = 0.154, p < 0.001) and SWB (r = 0.276, p < 0.001), while social activity was positively related to SWB (r = 0.191, p < 0.001). Based on the correlation analysis, we further conducted a chain mediation model analysis to explore the possible associations among these variables.

## Mediating effect analysis

The analysis of the chain mediation model included four multiple linear regressions, each accounting for covariates. Table 3 and Fig 2 present the results of the chain mediation analysis for this study. In the total effects model, SES was a significant predictor of residents' SWB (β = 1.583, p < 0.01). This relationship remained significant when class identity and social activity were also considered (β = 0.841, p < 0.05). Furthermore, SES has a significant effect on class identity (β = 0.289, p < 0.01) and social activity (β = 2.699, p < 0.01). Class identity and social activity were also significant predictors of SWB, with corresponding coefficients of 1.213 (p < 0.01) and 0.141 (p < 0.01). Class identity was a significant predictor of social activity, with a coefficient of 0.274 (p < 0.05). Using these path coefficients, the corresponding indirect effects were calculated. The indirect effects of SES on SWB, mediated by class identity and social activity, were 0.351 and 0.380, respectively. The chain mediation effect was 0.011, indicating that the effect of SES on SWB was transmitted between the mediating variables.

Finally, we used the bootstrap method to verify the total, direct, and indirect effects. As shown in Table 4, all the pathways were significant (p < 0.05). Specifically, the indirect effect of SES mediated by class identity on the SWB of residents was 0.351 (95% CI: 0.721, 1.587), whereas the indirect effect of SES on residents' SWB mediated by social activity was 0.380 (95% CI: 0.059, 0.240). The chain mediation effect, which was mediated by both class identity and social activity, was 0.011 (95% CI: 0.010, 0.093). The total indirect, direct, and total effects

**Table 2. The results of Pearson correlation analysis.**

| Variable | Socioeconomic status | Class identity | Social activities | Subjective well-being |
|---|---|---|---|---|
| Socioeconomic status | 1 | | | |
| Class identity | 0.130*** | 1 | | |
| Social activities | 0.486*** | 0.154*** | 1 | |
| Subjective well-being | 0.180*** | 0.276*** | 0.191*** | 1 |

Note

* p < 0.05

** p < 0.01

*** p < 0.001

**Table 3. Multiple stepwise regression results.**

| Variable | Subjective well-being | Class identity | Social activity | Subjective well-being |
|---|---|---|---|---|
| Socioeconomic status | 1.583*** (0.858, 2.308) | 0.289*** (0.178, 0.401) | 2.699*** (2.339, 3.059) | 0.841** (0.090, 1.592) |
| Class identity | | | 0.274*** (0.133, 0.415) | 1.213*** (0.948, 1.478) |
| Social activity | | | | 0.141*** (0.053, 0.228) |
| Age | 0.099*** (0.064, 0.133) | 0.014*** (0.009, 0.020) | -0.077*** (-0.095, -0.058) | 0.091*** (0.057, 0.125) |
| Gender | | | | |
| Female | | | | |
| Male | -0.124 (-1.127, 0.879) | -0.406*** (-0.571, -0.242) | -0.803*** (-1.316, -0.291) | 0.497 (-0.488, 1.482) |
| Ethnic | | | | |
| Non-Han | | | | |
| Han | 0.106 (-1.737, 1.949) | -0.275* (-0.593, 0.043) | -0.009 (-0.914, 0.896) | 0.451 (-1.355, 2.258) |
| Marriage | | | | |
| Married or have a spouse | | | | |
| Unmarried, divorced or widowed | 1.419** (0.226, 2.611) | 0.028 (-0.174, 0.230) | -0.181 (-0.765, 0.403) | 1.409** (0.256, 2.562) |
| Hukou | | | | |
| Rural | | | | |
| Urban | 0.228 (-0.923, 1.379) | 0.163* (-0.011, 0.338) | 1.210*** (0.636, 1.784) | -0.146 (-1.277, 0.984) |
| Health | 4.399*** (3.890, 4.907) | 0.339*** (0.249, 0.429) | 0.729*** (0.468, 0.990) | 3.872*** (3.371, 4.373) |
| Constant | 65.353*** (62.169, 68.538) | 2.799*** (2.238, 3.359) | 28.414*** (26.742, 30.086) | 57.854*** (53.700, 62.008) |
| Observations | 1926 | 1926 | 1926 | 1926 |
| R-Square | 0.171 | 0.065 | 0.311 | 0.212 |

Note
* $p < 0.1$
** $p < 0.05$
*** $p < 0.01$

of SES on SWB were 0.742 (95% CI: 0.891, 1.784), 0.841 (95% CI: 1.670, 3.148), and 1.583 (95% CI: 3.168, 4.281), respectively. All path effects were significant, suggesting the robustness of the mediation model results.

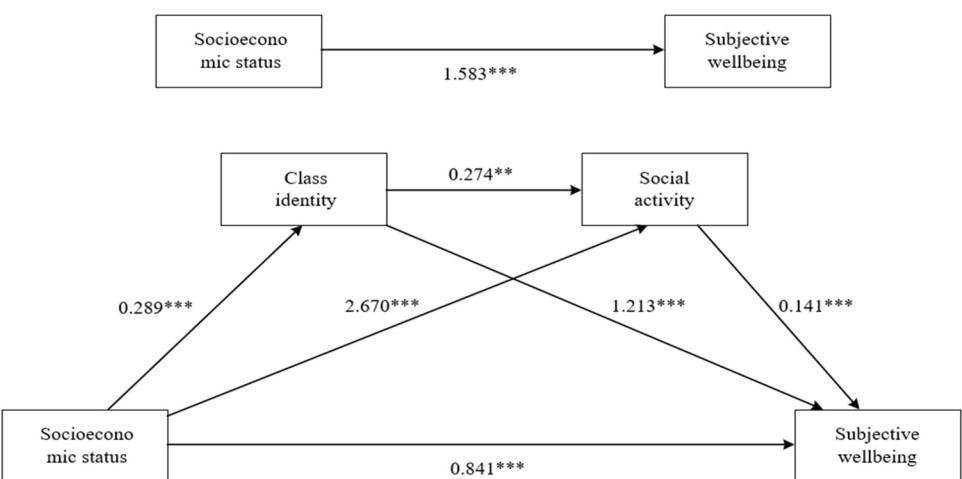

**Fig 2. Chain mediation model of socioeconomic status and subjective well-being through class identity and social activity.**

**Table 4. Bootstrap test results for multiple intermediary models.**

| Paths | Observed Coefficient | Bootstrap Standard Error | P | Lower | Upper |
|---|---|---|---|---|---|
| Indirect effect via class identity | 0.351 | 0.080 | 0.000 | 0.721 | 1.587 |
| Indirect effect via social activities | 0.380 | 0.123 | 0.002 | 0.059 | 0.240 |
| Indirect effect via class identity and social activities | 0.011 | 0.005 | 0.032 | 0.010 | 0.093 |
| Total indirect effect | 0.742 | 0.152 | 0.000 | 0.891 | 1.784 |
| Direct effect | 0.841 | 0.389 | 0.030 | 1.670 | 3.148 |
| Total effect | 1.583 | 0.374 | 0.000 | 3.168 | 4.281 |

Note: LLCI, lower level for confidence interval; ULCI, upper level for confidence interval.

## Discussion

In this study, we employed a chain mediation model to explore the relationship between SES, class identity, social activity participation, and SWB. As hypothesized, both class identity and social activity partially mediated the relationship between SES and SWB. Moreover, class identity and social activity participation exhibited a chain mediating effect in connecting SES with SWB. The robustness of the mediation analysis was confirmed through Bootstrap analyses.

The findings of this study supported both the direct effect (53.13%) and the indirect effect (46.87%) of SES on SWB, consistent with previous research [36, 37]. Several possible explanations could elaborate on the mechanism underlying this effect. Individuals with lower SES have experienced more negative emotions and stress, which may impair life satisfaction and SWB [38]. Moreover, lower SES was associated with unhealthy behaviors and limited leisure time, both of which can contribute to impaired SWB [39]. Conversely, favorable SES could facilitate access to more social support, educational resources, and material conditions, leading to increased personal resilience and coping capacity [40, 41]. Therefore, it is necessary to prioritize vulnerable groups with low SES and provide them with more material and spiritual support to enhance their SWB.

The result of the chain mediation model suggested that class identity served as a mediator between SES and SWB, accounting for 22.17% of the total effect. Class identity was rooted in socioeconomic resources and was a subjective mapping of SES [42]. Specifically, prior research suggested that higher SES is associated with greater political, cultural, and economic support, which potentially reinforces class identity [43]. Furthermore, class identity was influenced by subjective emotions, personality, values, and expectations, all of which exerted a consistent influence on SWB [44]. For instance, Pettit et al. found that class identity probably impacted SWB by shaping personal expectations and overall satisfaction [45]. Meanwhile, individuals with a stronger class identity were vulnerable to respect, recognition, and support, which positively impacted SWB through increased self-esteem, belonging, and social connectedness [46].

Similarly, the results from the mediation modeling showed that social activity participation mediated between SES and SWB, accounting for 24.01% of the total effect. Besides, the presence of a chain-mediated effect of class identity and social activity participation between SES and SWB was confirmed. Both SES and class identity are essential determinants of participation in social activities [47]. Scholars have found that individuals with higher SES usually possess better resources, social networks, and communication opportunities, which foster a supportive environment for participation in social activities [48, 49]. As stated in Bourdieu's theory of cultural capital, social class identity is associated with educational and cultural resources [50], contributing to a broader scope of activities and qualifications for participation. Furthermore, social activities play a crucial role in combating feelings of loneliness and social isolation [51]. It provided opportunities for positive interaction, emotional support, self-

fulfillment, and the development of intimate relationships. As an example, Zhang et al. found that positive feedback (e.g., praise, respect, and appreciation) received during social interactions positively influenced mental health and self-esteem [52]. Moreover, active participation in activities contributed not only to the development of a sense of belonging and social connectedness but also to social adaptation and psychological resilience [53]. All of these benefits could mitigate the negative impacts of SES and perceived class disadvantage, thereby contributing to increased SWB.

Based on the chain mediator model, we verified the conceptual framework and the three hypotheses. Notably, class identity and social activity had similar mediating effects, suggesting that both make meaningful contributions to how SES affects SWB (22.13% and 24.01%). These findings underscore the multifaceted dimension of SWB and the necessity to consider factors other than economic resources when addressing individual and community well-being. Incorporating social identity and participation in social activities into well-being improvement programs could lead to a more holistic and inclusive approach to improving SWB.

This study has several advantages. Firstly, as far as we know, this is the first study to examine the chain mediating effects of class identity and social activity participation on SES and SWB. This comprehensive exploration sheds light on the sequential mediation of the relationship between SES and SWB by class identity and social activity participation. Secondly, the study benefits from the use of well-represented and scientifically sound CGSS data, facilitating an examination of the mechanism between SES and SWB within a specific cultural context. Thirdly, by integrating class identity and participation in social activities as a mediator, this study provides a novel perspective on the psychological mechanisms through which SES influences SWB. However, the present study has several limitations. Firstly, this study adopts a cross-sectional design, possibly limiting the causal relationship between critical variables. A longitudinal design investigating causal effects could be conducted in future studies. Secondly, the mediating effect is approximately 46.87%, which suggests potentially unobserved roles and mechanisms. Future research should explore additional potential factors. Finally, the reliability of our results may be undermined due to potential bias in the self-reported data.

## Conclusion

Based on nationally representative data and the chain mediation model, this study investigated the mechanisms between SES, class identity, social activities, and SWB. The findings reveal a positive relationship between these four factors. Class identity and social activity partially mediated the effect of SES on SWB, and they had a chain mediating effect between SES and SWB. The results of this study have implications for improving residents' SWB. This positive mediation mechanism suggests that policymakers have the opportunity to enhance SWB in lower SES groups by promoting individual class identity and encouraging greater social activity participation. For instance, policymakers and practitioners can design empowerment programs that foster a sense of belonging and community while also providing opportunities for social engagement and support.

## Acknowledgments

We thank Chinese General Social Survey for their excellent work in database design and data collection and for allowing free access to the data.

## Author Contributions

**Conceptualization:** Baoqin Wang.

**Data curation:** Hang Zhao.

**Methodology:** Hang Zhao.

**Supervision:** Hao Shen, Yi Jiang.

**Writing – original draft:** Baoqin Wang, Hang Zhao.

**Writing – review & editing:** Baoqin Wang, Hang Zhao.

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
