## [Decision Letter · Decision Letter 0]

26 Jul 2023

PONE-D-23-18471Socioecnomic status and subjective well-being: The mediating and role of class identity and social activitiesPLOS ONE

Dear Dr. Jiang,

Thank you for submitting your manuscript to PLOS ONE. After careful consideration, we feel that it has merit but does not fully meet PLOS ONE’s publication criteria as it currently stands. Therefore, we invite you to submit a revised version of the manuscript that addresses the points raised during the review process.

The revised version should address all comments.

We look forward to receiving your revised manuscript.

Kind regards,

Petri Böckerman

Academic Editor

PLOS ONE

Journal Requirements:

Additional Editor Comments:

The revised version should address all comments.

Reviewers' comments:

Reviewer's Responses to Questions

**Comments to the Author**

1. Is the manuscript technically sound, and do the data support the conclusions?

Reviewer #1: No

Reviewer #2: No

Reviewer #3: Yes

2. Has the statistical analysis been performed appropriately and rigorously? 

Reviewer #1: No

Reviewer #2: Yes

Reviewer #3: Yes

3. Have the authors made all data underlying the findings in their manuscript fully available?

Reviewer #1: Yes

Reviewer #2: Yes

Reviewer #3: Yes

4. Is the manuscript presented in an intelligible fashion and written in standard English?

Reviewer #1: Yes

Reviewer #2: Yes

Reviewer #3: Yes

5. Review Comments to the Author

Reviewer #1: This is an interesting study that attempts to evaluate how much of the relationship between SES and subjective well-being is mediated by two psychosocial constructs: perceived social class identity and engagement in social activities.

Although the authors make the attempt to provide an important contribution to understanding the mechanisms behind SES on well-being, there are some gaps to be filled to be considered a technically sound scientific piece that supports the conclusions stated. In this line, some of the aspects concluded should be corroborated with some further analyses, considering some of the methodological limitations that this study has. Below, I provide some specific comments regarding these points:

- It is not clear what theory underlies the eventual relationship between SES, social class identity, and social activities. For instance, it sounds like authors are trying to connect intuitive relationships among variables rather than illustrate a causal theoretical framework that they want to test through a study. It would be ideal for the paper to frame a theory that clearly justifies the relationship among these three variables.

- The hypotheses proposed seem pertinent but basic and with low scientific risk. It would be recommended to state a stronger hypothesis according to the study and conclusions proposed.

- The fact that SES was measured in a subjective way and not using objective indicators (e.g., income level, neighborhood SES) is methodologically problematic. The way that SES is measured can be considered a measure of perceiver social status rather than SES. This is methodologically problematic as perceived social status and social class identity may be highly interrelated because they are part of the same measured construct. In that regard, social class identity could influence the perceived SES too.

- The discussion rephrases and expands on similar points already illustrated in the introduction. It would be expected to provide more discussion about the novelty of the analysis performed and how much of the effect of SES is mediated by these two mediators (class identity, social activity). From your analysis, only 1/3 of the total effect of perceived SES on SWB is explained by these two mediator variables. Why does this happen? What would explain the other 2/3? What this says about the effect of SES on SWB and if it is possible to mitigate this effect by targeting these two mediators?

- Lines 300 - 302: The conclusion stated is not necessarily supported by the analysis and the data analyzed. Sub-group and other types of analyses are required to elaborate such conclusions.

Reviewer #2: The authors aimed to investigate the impact of socioeconomic status, class identity, and social activity on Subjective well-being. There are some concerns about this study that need to be addressed.

Declarations:

1- Since this study was conducted on human samples, it is necessary to provide necessary and sufficient explanations in the Ethics Statement.

Introduction

2- The objectives and hypotheses of the study should be written as a continuous text and not as separate cases at the end of the Introduction section.

Methods:

3- The authors should either describe the sampling method completely and in detail or provide the necessary reference to a valid study in this regard.

4- Limited information has been provided regarding the development of SWBS-CC and its subscales and validating the scale.

5- Measuring socioeconomic status and class identity, which are two complex indicators, is almost impossible with only two questions, and the validity of the results is highly questionable.

6- Limited information has been provided regarding the development of social activity questionnaire and its subscales and validating the scale.

7- What do the authors mean by the word “residents” at the first paragraph of the Discussion section.

Reviewer #3: The study examines the relationship between subjective well-being and socioeconomic status, class identity and participation in social activities. They found that subjective well-being correlates significantly to these other factors. Additionally, they found that the effect of socioeconomic status on subjective well-being was mediated by social activity participation and class identity.

Please see below comments and suggestions based on my review:

1. It is not clear what the effect size measures and the 95% CI reported in the study represents.

2. Page 3, Line 43 - SWB should be written in full (sentences should not start with acronym). Other parts of the manuscript should be corrected as well.

3. Page 3, Line 50 - 52. The justification does not flow with the previous statement. Consider deleting or reframing.

4. Page 3, Line 53. is instead of was and for instead of in.

5. Page 4, Line 63. depends instead of depended

6. Page 4, Line 72. delete the and change relationship to correlation (correlation is more specific)

7. Page 6, Line 111 - 112. "After excluding observations with important missing data..."

8. Page 6, Line 114. SWB - see comment 2 above

9. Page 7, Line 125 - 126. What were the options on the Likert scale?

10. Page 8, Line 147. The word "illiterate" may not be appropriate in this scenario. Consider using "no formal education".

11. Page 8, Line 152. Frequency instead of number

12. Page 8, Line 153. Were the assumptions for Pearson's correlation met?

13. Page 9, Line 168. Illiteracy - see comment 10 above.

14. Page 9, Line 170. The mean age not mean scores for age

15. Table 1 (Age (years))

16. Table 1 (illiterate - see comment 10)

17. Page 11, Line 197 - 198. The statement on confirmation of previously stated hypotheses should be moved to discussion

18. Page 15. The first paragraph of the discussion is verbose. I suggest providing a more succinct summary of the results here.

Thank you.

6. PLOS authors have the option to publish the peer review history of their article (what does this mean?). If published, this will include your full peer review and any attached files.

Reviewer #1: No

Reviewer #2: **Yes: **Yaser Sarikhani

Reviewer #3: No

---

## [Author Response · Author response to Decision Letter 0]

12 Aug 2023

Dear Editor and Reviewers,

Thank you for offering us an opportunity to improve the quality of our submitted manuscript. We appreciated very much the reviewers’ constructive and insightful comments. These comments are essential to improving the rigor and scientific quality of the manuscript. We have taken all these comments and suggestions into account and have carefully revised this manuscript. Changes are shown in red in the revised version. Appended to this letter is our point-by-point response to the comments raised by the reviewers.

Kind regards,

Yi Jiang

Responds to the reviewers' comments

Reviewer 1

1. It is not clear what theory underlies the eventual relationship between SES, social class identity, and social activities. For instance, it sounds like authors are trying to connect intuitive relationships among variables rather than illustrate a causal theoretical framework that they want to test through a study. It would be ideal for the paper to frame a theory that clearly justifies the relationship among these three variables. 

Response: We sincerely appreciate the valuable advice from the reviewer. We attempt to elucidate the potential association between socioeconomic status, social class identity, social activities, and subjective well-being through the relevant literature. In this way, we proposed our research framework and hypothesized potential pathways. Although we have referenced the literature based on certain theories, we recognize the need to complement and refine these theories to bolster the rationality and scientific validity of this manuscript. We conducted extensive inquiries and found several theoretical bases to support our research hypotheses. Specifically, we employed stress theory, Bourdieu's theory of cultural capital, and social identity theory to elaborate on the associations between socioeconomic status, class identity, and participation in social activities. The relationship between socioeconomic status, class identity, social activity participation, and subjective well-being was elaborated using stress theory, social comparison theory, and social support theory. Furthermore, we have adjusted some descriptions and expressions based on the previous version. The detailed modifications are as follows:

Socioeconomic status (SES) is an essential influencing factor for SWB. Research has shown that SES could contribute to SWB through various external and internal mechanisms. Stress theory could elaborate on the mechanisms and pathways through which SES can influence well-being, especially the concepts of coping resources and stressors. Individuals with higher SES tended to enjoy better life circumstances and more social resources, buffering the effects of adverse events [6, 7]. Furthermore, individuals with higher SES were exposed less to stressful and uncontrollable life events, which were associated with higher SWB [8]. Subjective class identity is an individual's perception of their position in the social class structure, reflecting the individual's internalization, adaptation, and acceptance of the social environment and group [9]. Class identity depends heavily on the degree of individual demand fulfillment. Higher demand fulfillment was associated with higher SWB [10]. Social comparison theory suggests that people determine their values and the realization of their demands by comparing themselves to others [11]. Therefore, the SWB of individuals depended not only on their SES and class identity but also on their perceptions of their position relative to others. Similarly, engaging in social activities played an important role in the well-being of the population. Social support theory emphasizes the importance of social relationships and the support individuals receive from their social networks [12]. Participation in social activities was associated with more active social networks [13, 14]. These social networks provided emotional, informational, and instrumental support that facilitated the reduction of isolation and stress and increased resilience and SWB [15].

Abundant evidence has demonstrated a positive correlation between SES and social class identity. Bourdieu's theory of cultural capital emphasizes the importance of cultural capital in shaping social class identity [16, 17], such as education, cultural resources, and social skills. However, cultural capital acquisition is often influenced by SES [18]. For example, individuals from higher SES backgrounds were more likely to have access to high-quality education, exposure to arts and cultural experiences, and opportunities to develop cultural tastes and skills [19,20]. The social identity theory also explained the association between SES and class identity. Social identity theory suggests that people tend to categorize themselves based on various social dimensions, especially SES [21]. Individuals with similar SES often have shared experiences and challenges, contributing to the development of a common class identity. Research has shown the significant and positive impact of SES and class identity on individuals' engagement in social activities [22]. Based on stress theory, higher SES and class identity were associated with more social support and fewer stressors, possibly promoting individuals to engage more in social activities [23]. Individuals with higher SES and class identity typically had access to a wider range of social support and resources, which could facilitate participation in social activities [24, 25]. (Line 57-96)

2. The hypotheses proposed seem pertinent but basic and with low scientific risk. It would be recommended to state a stronger hypothesis according to the study and conclusions proposed.

Response: Thank you for your helpful comments. Taking your comments into account, we have adjusted the assumptions. We emphasized the main conclusions and framework of the article in our hypothesis. The details are as follows:

In conclusion, although the effect of SES on SWB has been explored, few previous scholars have examined the mediating role of class identity and social activities. Therefore, we used the chain mediation model to examine the potential relationship among SES, class identity, social activity, and SWB. The above theories and literature have revealed the interconnections among SES, class identity, and social activities, emphasizing their importance as influences on SWB. Based on these, the conceptual framework (shown in Figure 1) and the following hypotheses were proposed: 1) there is a significant association between SES, class identity, participation in social activities, and SWB; 2) class identity and participation in social activities mediate between SES and SWB; 3) class identity and social activity participation have a chain-mediated role

between SES and SWB. (Line 97-107)

Fig 1. The conceptual model based on previous research and theory.

3. The fact that SES was measured in a subjective way and not using objective indicators (e.g., income level, neighborhood SES) is methodologically problematic. The way that SES is measured can be considered a measure of perceiver social status rather than SES. This is methodologically problematic as perceived social status and social class identity may be highly interrelated because they are part of the same measured construct. In that regard, social class identity could influence the perceived SES too.

Response: We sincerely appreciate your valuable comments, as they significantly enhance the scientific validity and rigor of our article. We clearly recognize the problem in measurement, ignoring the similarity between subjective socioeconomic status and subjective class identity. In response, we conducted an extensive literature search and ultimately adopted a composite measure of education and income as an indicator of socioeconomic status. Consequently, we reperformed the relevant analyses, and the corresponding results were updated in the revised manuscript. The specific methodology is elucidated in the paper as follows:

The present study employed SES as a key variable. Following relevant studies, we employed education and income as measures of SES [29,30,31]. Education was categorized as follows: (1) uneducated, (2) primary school and below, (3) middle school, (4) high school, and (5) college and higher. Personal income was stratified into quintiles (1-5) based on annual earnings [32]. To create the SES composite score, standardized z-scores for education and income were averaged. (Line 141-146)

Furthermore, we adjusted the description of the control variables as education became a SES indicator. The specifics are as follows:

We included several control variables to improve the estimation accuracy, including age, gender, ethnicity, marriage, household registration, and self-rated health. Ethnicity was divided into Han and others, and household registration was divided into urban and rural. Marital status was classified as: unmarried, divorced, and widowed; married and having a spouse. The self-assessment of health is based on the question, "How do you feel about your current state of health?", with values from 1 (very poor) to 5 (very good). (Line 164-170)

4. The discussion rephrases and expands on similar points already illustrated in the introduction. It would be expected to provide more discussion about the novelty of the analysis performed and how much of the effect of SES is mediated by these two mediators (class identity, social activity). From your analysis, only 1/3 of the total effect of perceived SES on SWB is explained by these two mediator variables. Why does this happen? What would explain the other 2/3? What this says about the effect of SES on SWB and if it is possible to mitigate this effect by targeting these two mediators?

Response: We honestly appreciate your comments and questions. The corresponding changes and responses are listed below:

(1) We have added the strengths of the article in the discussion section to illustrate the innovation of this study. The details are shown as follows:

This study has several advantages. Firstly, as far as we know, this is the first study to examine the chain-mediated effects of class identity and social activity participation on SES and SWB. This comprehensive exploration sheds light on the sequential mediation of the relationship between SES and SWB by class identity and social activity participation. Secondly, the study benefits from the use of well-represented and scientifically sound CGSS data, facilitating an examination of the mechanism between SES and SWB in the specific cultural context of China. Thirdly, by integrating class identity and participation in social activities as mediators, this study provides a novel perspective on the psychological mechanisms through which SES influences SWB. However, the present study has several limitations. Firstly, this study adopts a cross-sectional design, possibly limiting the causal relationship between critical variables. A longitudinal design investigating causal effects could be conducted in future studies. Secondly, the mediating effect is approximately 46.87%, which suggests potentially unobserved roles and mechanisms. Future research should explore additional potential factors. Finally, the reliability of our results may be undermined due to potential bias in the self-reported data. (Line 296-311)

(2) We have adjusted the interpretation of mediating roles in our discussion and emphasized the extent to which the effect of socioeconomic status on subjective class identity is influenced by class identity and social activity. The details are provided as follows:

The findings of this study supported both the direct effect (53.13%) and the indirect effect (46.87%) of SES on SWB, consistent with previous research [36, 37]. Several possible explanations could elaborate on the mechanism underlying this effect. Individuals with lower SES have experienced more negative emotions and stress, which may impair life satisfaction and SWB [38]. Moreover, lower SES was associated with unhealthy behaviors and limited leisure time, both of which can contribute to impaired SWB [39]. Conversely, favorable SES could facilitate access to more social support, educational resources, and material conditions, leading to increased personal resilience and coping capacity [40, 41]. Therefore, it is necessary to prioritize vulnerable groups with low SES and provide them with more material and spiritual support to enhance their SWB. 

The result of the chain mediation model suggested that class identity served as a mediator between SES and SWB, accounting for 22.17% of the total effect. Class identity was rooted in socioeconomic resources and was a subjective mapping of SES [42]. Specifically, prior research suggested that higher SES is associated with greater political, cultural, and economic support, which potentially reinforces class identity [43]. Furthermore, class identity was influenced by subjective emotions, personality, values, and expectations, all of which exerted a consistent influence on SWB [44]. For instance, Pettit et al. found that class identity probably impacted SWB by shaping personal expectations and overall satisfaction [45]. Meanwhile, individuals with a stronger class identity were vulnerable to respect, recognition, and support, which positively impacted SWB through increased self-esteem, belonging, and social connectedness [46].

Similarly, the results from the mediation modeling showed that social activity participation mediated between SES and SWB, accounting for 24.01% of the total effect. Besides, the presence of a chain-mediated effect of class identity and social activity participation between SES and SWB was confirmed. Both SES and class identity are essential determinants of participation in social activities [47]. Scholars have found that individuals with higher SES usually possess better resources, social networks, and communication opportunities, which foster a supportive environment for participation in social activities [48, 49]. As stated in Bourdieu's theory of cultural capital, social class identity is associated with educational and cultural resources [50], contributing to a broader scope of activities and qualifications for participation. Furthermore, social activities play a crucial role in combating feelings of loneliness and social isolation [51]. It provided opportunities for positive interaction, emotional support, self-fulfillment, and the development of intimate relationships. As an example, Zhang et al. found that positive feedback (e.g., praise, respect, and appreciation) received during social interactions positively influenced mental health and self-esteem [52]. Moreover, active participation in activities contributed not only to the development of a sense of belonging and social connectedness but also to social adaptation and psychological resilience [53]. All of these benefits could mitigate the negative impacts of SES and perceived class disadvantage, thereby contributing to increased SWB.

Notably, class identity and social activity had similar mediating effects, suggesting that both make meaningful contributions to how SES affects SWB (22.13% and 24.01%). These findings underscore the multifaceted dimension of SWB and the necessity to consider factors other than economic resources when addressing individual and community well-being. Incorporating social identity and participation in social activities into well-being improvement programs could lead to a more holistic and inclusive approach to improving SWB. (Line 245-295)

(3) Since we changed the measure of socioeconomic status, the interpretation of the relationship between socioeconomic status and subjective well-being by the two mediating variables (class identity and social activity participation) has been altered. 

The study revealed that about 47% of the effect of socioeconomic status on subjective well-being was mediated by class identity and social activity, indicating that other mediating factors or confounding variables were not accounted for in the analysis. For instance, unexplained portions of the effect (about 53%) might be influenced by factors such as personal values, cultural differences, living environment, job satisfaction, social support, and coping mechanisms. Limited by data accessibility and analytical methods, we were unable to include all potential factors for analysis to increase the level of model explanation. Therefore, future studies could explore more potential pathways to complement the mechanisms of the effects of socioeconomic status on subjective well-being. 

(4) Our findings suggest that class identity and social activity participation, as important mediating variables, can partially mediate the effect of SES on SWB. This positive mediating mechanism suggests that policymakers can improve the SWB of vulnerable groups by increasing individuals' class identity and participation in social activities. For example, policymakers and practitioners can design programs that empower individuals, foster a sense of belonging and community, and provide opportunities for social engagement and support.

5. Lines 300 - 302: The conclusion stated is not necessarily supported by the analysis and the data analyzed. Sub-group and other types of analyses are required to elaborate such conclusions.

Response: We appreciate you pointing out the problems. We have adjusted the description in the conclusion section to accurately present the findings and suggestions. The specific modifications are described as follows:

Based on nationally representative data and a chained mediation model, this study investigated the mechanisms between SES, class identity, social activity, and SWB. The findings reveal a positive relationship between these four factors. Class identity and social activity partially mediated the effects of SES on SWB, and they had a chain mediating effect between SES and SWB. This positive mediation mechanism suggests that policymakers have the opportunity to enhance SWB in lower SES groups by promoting individual class identity and encouraging greater social activity participation. For instance, policymakers and practitioners can design empowerment programs that empower individuals, foster a sense of belonging and community, and provide opportunities for social engagement and support. (Line 313-323)

Reviewer 2

1. Since this study was conducted on human samples, it is necessary to provide necessary and sufficient explanations in the Ethics Statement.

Response: Thank you for pointing out our shortcomings here. The CGSS was conducted in accordance with the ethical principles of the Declaration of Helsinki. The Ethics Committees of Renmin University of China and Hong Kong University of Science and Technology are responsible for ethical approval and consent to participate. After ensuring that the information was understood by the respondents, each potential respondent was fully informed about the purpose of the study, the methodology, the source of funding, any possible conflicts of interest, any discomfort that the study might cause, and any other information relevant to the study. We received authorization to use the publicly accessible CGSS. This study was conducted based on de-identified publicly available CGSS (http://cgss.ruc.edu.cn/) data. Therefore, ethical approval or informed consent was no longer required for this study.

2. The objectives and hypotheses of the study should be written as a continuous text and not as separate cases at the end of the Introduction section.

Response: Thank you for your helpful comments. Taking your comments into account, we have adjusted the assumptions. The modifications are as follows:

In conclusion, although the effect of SES on SWB has been explored, few previous scholars have examined the mediating role of class identity and social activities. Therefore, we used the chain mediation model to examine the potential relationship among SES, class identity, social activity, and SWB. The above theories and literature have revealed the interconnections among SES, class identity, and social activities, emphasizing their importance as influences on SWB. Based on these, the conceptual framework (shown in Figure 1) and the following hypotheses were proposed: 1) there is a significant association between SES, class identity, participation in social activities, and SWB; 2) class identity and participation in social activities mediate between SES and SWB; 3) class identity and social activity participation have a chain-mediated role between SES and SWB. (Line 97-107)

Fig 1. The conceptual model based on previous research and theory.

3.The authors should either describe the sampling method completely and in detail or provide the necessary reference to a valid study in this regard.

Response: Thank reviewer 2 for the critical comments. The Chinese General Social Survey (CGSS) is a large-scale nationwide survey program that covers both urban and rural residents nationwide. The survey was conducted using stratified three-stage probability sampling. Depending on the sampling stratum, the sampling units at each stage are slightly different, as shown in Table 1. Specifically, the survey population is divided into two main categories: the first is the mandatory layer, which consists of households in the municipal districts of the selected large cities; the second is the lottery layer, which consists of all households in the country excluding the mandatory layer of municipal districts. Information obtained from: http://cgss.ruc.edu.cn/xmwd/cysj.htm

Table 1 Sampling units for each stage

 First-stage sampling units Second-stage sampling units Third-stage sampling units

Mandatory layer Street Neighborhood councils Households

Lottery Layer Districts, County-level cities, Counties Neighborhood councils, Village councils Households

4.Limited information has been provided regarding the development of SWBS-CC and its subscales and validating the scale.

Response: Thank Reviewer 2 for the critical comments. In the study, subjective well-being was measured with the Subjective Well-being Scale for Chinese Citizens (SWBS-CC) by scholar Xing Zhanjun from the Chinese Mainland. With a reliability of 0.838 and a validity of 0.864, the scale is an effective measurement in research on urban citizens in the Chinese Mainland. The scale was based on a six-point Likert scale, with values ranging from 1 (completely disagree) to 6 (completely agree). It consists of 20 items (10 dimensions), including satisfaction and abundance, mental health, confidence in society, growth and progress, goals and personal values, self-acceptance, physical health, psychological balance, adjustment to interpersonal relationships, and family climate. Total scores range from 20 to 120, with higher scores indicating higher subjective well-being. The DOI numbers of the references are as follows: doi:10.3389/fpsyg.2022.1050486 and doi:10.1016/j.jad.2022.01.089.

5.Measuring socioeconomic status and class identity, which are two complex indicators, is almost impossible with only two questions, and the validity of the results is highly questionable.

Response: We sincerely appreciate the valuable advice from the reviewer 2. Based on the suggestions made by you and reviewer 1, and after careful consideration, we have decided to make the following changes:

The present study employed SES as a key variable. Following relevant studies, we employed education and income as measures of SES [29,30,31]. Education was categorized as follows: (1) uneducated, (2) primary school and below, (3) middle school, (4) high school, and (5) college and higher. Personal income was stratified into quintiles (1-5) based on annual earnings [32]. To create the SES composite score, standardized z-scores for education and income were averaged. 

According to social identity theory, social identity derives primarily from group membership or qualifications and is rooted in people's subjective perceptions of their class identity. To measure class identity, self-rated class identity was used based on previous research (doi:10.3389/fpsyg.2021.627610). The CGSS asked the question about class identity: "In general, where do you personally stand in the current society?" Responses were recorded on a 10-point Likert scale, with higher scores indicating higher levels of class identity. (Line 141-152)

We have adjusted these descriptions to support the plausibility and scientific validity of this manuscript. Additionally, we performed the data analysis again due to a change in the way the measurements were taken. All results are updated in the revised manuscript.

6.Limited information has been provided regarding the development of social activity questionnaire and its subscales and validating the scale.

Response: Thank Reviewer 2 for the critical comments. In this study, we referred to several studies to measure social activities. Social activity is measured by the question "How often in the past year have you engaged in the following activities". Answers included watching television, going to the movies, shopping, reading, going to shows, meeting with relatives, meeting with friends, listening to music, exercising, watching sports, doing handicrafts, and surfing the Internet. Answers to each question ranged from 1 to 5. 1 = never, 2 = several times a year, 3 = several times a month, 4 = several times a week, and 5 = daily. In addition, reliability and validity analyses were conducted to determine the reliability and validity of the social activity measures in this study. The results of reliability (Cronbach's alpha = 0.674) and validity (KMO = 0.782) showed that the questions responded well to social activities. 

The DOI numbers for the references are doi:10.3389/fpubh.2022.967170, doi:10.1016/j.jad.2022.09.031, and doi:10.1007/s40520-021-02036-1.

7.What do the authors mean by the word “residents” at the first paragraph of the Discussion section.

Response: Thank you for pointing this out. The word refers to individuals in the study population. We recognize that the use of this term is inappropriate and have changed it to “individual”.

Reviewer 3

1.It is not clear what the effect size measures and the 95% CI reported in the study represents.

Response: In regression analysis, effect sizes and 95% confidence intervals are important statistical measures that help you understand the strength and precision of the relationship between variables. Effect sizes quantify the strength and direction of the relationship between variables. For example, the total effect of SES on SWB in this study was 1.583. This suggests that one unit change in socioeconomic status leads to an average change in SWB of 1.583 units in the same direction, keeping all other variables unchanged. However, we use unstandardized coefficients and cannot directly compare the magnitude of individual path coefficients because these variables have different magnitudes, ranges of values, and degrees of variability. The 95% confidence interval (CI) provides a range within which you are reasonably confident that the true population parameter lies. The 95% CI indicates that if we were to repeat the study multiple times and compute CIs for each sample, about 95% of those CIs would contain the true population parameter. For example, a regression analysis yields an effect size of β = 1.583 with a 95% CI of [0.858, 2.308], which means that we are 95% sure that the true effect size is between 0.858 and 2.308. A wider CI indicates greater uncertainty, while a narrower CI indicates a more precise estimate.

2.Page 3, Line 43 - SWB should be written in full (sentences should not start with acronym). Other parts of the manuscript should be corrected as well.

Response: We appreciate Reviewer 3 pointing this out. We apologize for our carelessness. We have completed the changes at the appropriate places in the manuscript. (Line 48-49)

3.Page 3, Line 50 - 52. The justification does not flow with the previous statement. Consider deleting or reframing.

Response: Thank reviewer 3 for the critical comments. The previous description was intended to emphasize the importance of subjective well-being. The last sentence is intended to emphasize the importance of research on subjective well-being and the factors that influence it. We adjusted the description appropriately and changed it to “Therefore, it is necessary to extensively explore the factors and potential pathways influencing SWB.” (Line 55-56)

4.Page 3, Line 53. is instead of was and for instead of in.

Response: We were really sorry for our careless mistakes. Thank you for your reminder. We have corrected the “was” into “is” .and “in” into “for” (Line 57)

5.Page 4, Line 63. depends instead of depended

Response: We sincerely thank the reviewer for careful reading. As suggested by the reviewer, we have corrected the “depended” into “depends”. (Line 67)

6.Page 4, Line 72. delete the and change relationship to correlation (correlation is more specific)

Response: Thank reviewer 3 for the useful comments. We have made changes at the correct places in the text. (Line 79)

7.Page 6, Line 111 - 112. "After excluding observations with important missing data..."

Response: Thank reviewer 3 for the useful comments. We have used your suggestions to make changes in the text. (Line 128)

8.Page 6, Line 114. SWB - see comment 2 above

Response: We apologize for our carelessness. We have completed the changes at the appropriate places in the manuscript.

9. Page 7, Line 125 - 126. What were the options on the Likert scale?

Response: Thank you for your question. The options here included 1 (very poor), 2 (poor), 3 (fair), 4 (good), and 5 (very good). Because Reviewer 1 and Reviewer 2 questioned the subjective measure of socioeconomic status, we used education and income as measures of socioeconomic status. We have taken your comments and the approach of the relevant literature and re-described the measurement. Details of the changes are as follows:

The present study employed SES as a key variable. Following relevant studies, we employed education and income as measures of SES [29,30,31]. Education was categorized as follows: (1) uneducated, (2) primary school and below, (3) middle school, (4) high school, and (5) college and higher. Personal income was stratified into quintiles (1-5) based on annual earnings [32]. To create the SES composite score, standardized z-scores for education and income were averaged. (Line 141-146)

10.Page 8, Line 147. The word "illiterate" may not be appropriate in this scenario. Consider using "no formal education".

Response: Thanks to Reviewer 3 for careful checks. We are sorry for our carelessness. Based on your comments, we have made the corrections to make the word harmonized within the whole manuscript. We have changed illiterate to uneducated. (Line 143)

11.Page 8, Line 152. Frequency instead of number

Response: We feel sorry for our carelessness. In our resubmitted manuscript, it is revised. Thanks for your correction. (Line 175)

12.Page 8, Line 153. Were the assumptions for Pearson's correlation met?

Response: The Pearson test requires that the data meet normality or asymptotic normality. Related studies have shown that a distribution is called approximate normal if skewness or kurtosis of the data are between − 1 and + 1 (DOI: 10.4103/aca.ACA_157_18). Therefore, we calculated the skewness and kurtosis of socioeconomic status, class identity, social activity, and subjective well-being using STATA. All skewnesses and kurtosis are less than 1, as shown in Table 2. Accordingly, we hold that the Pearson test is satisfied. 

Table 2 Skewness and kurtosis of the variables

Variable Skewness Kurtosis

Socioeconomic status 0.526 0.000

Class identity 0.006 0.015

Social activity 0.597 0.000

Subjective well-being 0.000 0.154

13.Page 9, Line 168. Illiteracy - see comment 10 above.

Response: Thanks for your careful checks. We are sorry for our carelessness. Based on your comments, we have made the corrections to make the word harmonized within the whole manuscript.

14.Page 9, Line 170. The mean age not mean scores for age

Response: We are very grateful for the comments made by Review 3, which we have replaced with the following: “The mean of age, SES, class identity, social activity, and SWB were 49.43, 0.00, 4.35, 28.22, and 86.99, respectively.” (Line 191-193)

15.Table 1 (Age (years))

Response: Thanks for your correction. In our resubmitted manuscript, it is revised. 

16.Table 1 (illiterate - see comment 10)

Response: Thanks for your careful checks. We included education as one of the indicators of socioeconomic status and added self-rated health as a covariate. Therefore, Table 1 is not showing education.

As a result of changing the SES measurements, we re-performed the data analysis and updated the relevant results in the revised manuscript.

17.Page 11, Line 197 - 198. The statement on confirmation of previously stated hypotheses should be moved to discussion

Response: We are very appreciative of the comments made by Review 3 and we have removed this statement and moved it to the discussion section.

18.Page 15. The first paragraph of the discussion is verbose. I suggest providing a more succinct summary of the results here.

Response: We are very grateful for Review 3's comments, and we have revised the paragraph as follows:

In this study, we employed a chain mediation model to explore the relationship between SES, class identity, social activity participation, and SWB. As hypothesized, both class identity and social activity partially mediated the relationship between SES and SWB. Moreover, class identity and social activity participation exhibited a chain mediating effect in connecting SES with SWB. The robustness of the mediation analysis was confirmed through Bootstrap analyses. (Line 239-244)

---

## [Decision Letter · Decision Letter 1]

21 Aug 2023

PONE-D-23-18471R1Socioeconomic status and subjective well-being: The mediating role of class identity and social activitiesPLOS ONE

Dear Dr. Jiang,

Thank you for submitting your manuscript to PLOS ONE. After careful consideration, we feel that it has merit but does not fully meet PLOS ONE’s publication criteria as it currently stands. Therefore, we invite you to submit a revised version of the manuscript that addresses the points raised during the review process. The final version should address the remaining concerns.

We look forward to receiving your revised manuscript.

Kind regards,

Petri Böckerman

Academic Editor

PLOS ONE

Journal Requirements:

Reviewers' comments:

Reviewer's Responses to Questions

**Comments to the Author**

1. If the authors have adequately addressed your comments raised in a previous round of review and you feel that this manuscript is now acceptable for publication, you may indicate that here to bypass the “Comments to the Author” section, enter your conflict of interest statement in the “Confidential to Editor” section, and submit your "Accept" recommendation.

Reviewer #2: All comments have been addressed

Reviewer #3: (No Response)

2. Is the manuscript technically sound, and do the data support the conclusions?

Reviewer #2: Yes

Reviewer #3: Yes

3. Has the statistical analysis been performed appropriately and rigorously? 

Reviewer #2: Yes

Reviewer #3: Yes

4. Have the authors made all data underlying the findings in their manuscript fully available?

Reviewer #2: Yes

Reviewer #3: Yes

5. Is the manuscript presented in an intelligible fashion and written in standard English?

Reviewer #2: Yes

Reviewer #3: Yes

6. Review Comments to the Author

Reviewer #2: Based on the changes made to the Manuscript by the authors, it appears that the text has undergone an acceptable upgrade. Therefore, there is no more comment in this regard.

Reviewer #3: Thank you for your previous detailed responses to my comments. While I appreciate the explanation to point one, it is still unclear to a reader what those numbers represent. It is not immediately clear in the abstract and results that this was a multivariate linear regression analysis. The authors can consider reporting as: "For each unit increase in socioeconomic status score, the subjective well-being score was expected to increase by 0.351 (95% CI 0.721, 1.587)..."

The indirect effect of

socioeconomic status on subjective well-being mediated by class identity was 0.351

(95% CI: 0.721, 1.587)

7. PLOS authors have the option to publish the peer review history of their article (what does this mean?). If published, this will include your full peer review and any attached files.

Reviewer #2: **Yes: **Yaser Sarikhani

Reviewer #3: No

---

## [Author Response · Author response to Decision Letter 1]

23 Aug 2023

Dear Editor,

We appreciate your reply. We apologize for the incorrect formatting of the references due to our mistake. We have adjusted the reference format to ensure it is complete and correct. And all of the cited literature is published. Appended to this letter is our point-by-point response to the comments raised by the reviewers. Changes are shown in red in the revised version.

Kind regards,

Yi Jiang

Responds to the reviewers' comments

Reviewer 2

1 Based on the changes made to the Manuscript by the authors, it appears that the text has undergone an acceptable upgrade. Therefore, there is no more comment in this regard. 

Response: Thank you very much for your valuable suggestions and comments.

Reviewer 3

1 Thank you for your previous detailed responses to my comments. While I appreciate the explanation to point one, it is still unclear to a reader what those numbers represent. It is not immediately clear in the abstract and results that this was a multivariate linear regression analysis. The authors can consider reporting as: "For each unit increase in socioeconomic status score, the subjective well-being score was expected to increase by 0.351 (95% CI 0.721, 1.587)..."

Response: We appreciate your comments very much. We apologize for not clearly showing that the chain mediation analysis is based on several multiple linear regressions (as shown in Table 3). We have adapted pertinent elucidations within the manuscript to explicitly expound that the chain-mediated model comprises the four multiple linear regressions presented in Table 3. Furthermore, we utilized Figure 2 to present the connection between socioeconomic status, class identity, participation in social activities, and subjective well-being more visually. Regarding the query that the various coefficients (β) and p values we report may be difficult for the reader to understand, we inquired about how these coefficients are reported in the relevant literature. We found that the presentation of the chain-mediated modeling results in the manuscript is consistent with other relevant literature (DOI numbers are displayed at the end), expressed as the effect of A on B (coefficients and p-values, or 95% CI). This mode of presentation indicates the direction, degree, and precision of the connection between the two-by-two variables, which may not affect the reader's understanding of the results. The reader can interpret the coefficients (β) and p-values as reported by the multiple linear regression. For example, "socioeconomic status was a significant predictor of individuals' subjective well-being (β = 1.583, p < 0.01)" can be interpreted as "for every one-unit increase in socioeconomic status score, the subjective well-being score was expected to increase by 1.583 (p < 0.01)." Moreover, we believe that this presentation is concise and clear, thereby reducing the reading burden on the reader.

The various indirect effects are products of the corresponding path coefficients. The magnitude of the indirect effect indicates how much of the total effect between the independent variable (X) and the dependent variable (Y) is explained by the mediating variable (M). The significance, along with the 95% confidence interval, of an indirect effect determines whether the observed indirect effect may have occurred at random. Furthermore, 95% confidence intervals for indirect, direct, and total effects were tested using the Bootstrap method to ensure that these findings were not coincidental (as shown in Table 4). Therefore, the interpretation of the magnitude of the indirect effect may differ from the preceding regression coefficients. 

The details of the modifications are shown as follows: 

The analysis of the chain mediation model included four multiple linear regressions, each accounting for covariates. Table 3 and Figure 2 present the results of the chain mediation analysis for this study. In the total effects model, SES was a significant predictor of individuals' SWB (β = 1.583, p < 0.01). This relationship remained significant when class identity and social activity were also considered (β = 0.841, p < 0.05). Furthermore, SES has a significant effect on class identity (β = 0.289, p < 0.01) and social activity (β = 2.699, p < 0.01). Class identity and social activity were also significant predictors of SWB, with corresponding coefficients of 1.213 (p < 0.01) and 0.141 (p < 0.01). Class identity was a significant predictor of social activity, with a coefficient of 0.274 (p < 0.05). Using these path coefficients, the corresponding indirect effects were calculated. The indirect effects of SES on SWB, mediated by class identity and social activity, were 0.351 and 0.380, respectively. The chain mediation effect was 0.011, indicating that the effect of SES on SWB was transmitted between the mediating variables. (Line 209-222)

The DOI numbers of the references for the reporting style are specified below: 

(1) 10.1371/journal.pone.0122128

(2) 10.1371/journal.pone.0231628

(3) 10.1371/journal.pone.0280701

(4) 10.1186/s12889-022-14711-7

(5) 10.1371/journal.pone.0289092

---

## [Decision Letter · Decision Letter 2]

29 Aug 2023

Socioeconomic status and subjective well-being: The mediating role of class identity and social activities

PONE-D-23-18471R2

Dear Dr. Jiang,

We’re pleased to inform you that your manuscript has been judged scientifically suitable for publication and will be formally accepted for publication once it meets all outstanding technical requirements.

Kind regards,

Petri Böckerman

Academic Editor

PLOS ONE

Additional Editor Comments (optional):

Reviewers' comments:

Reviewer's Responses to Questions

**Comments to the Author**

1. If the authors have adequately addressed your comments raised in a previous round of review and you feel that this manuscript is now acceptable for publication, you may indicate that here to bypass the “Comments to the Author” section, enter your conflict of interest statement in the “Confidential to Editor” section, and submit your "Accept" recommendation.

Reviewer #2: All comments have been addressed

Reviewer #3: All comments have been addressed

2. Is the manuscript technically sound, and do the data support the conclusions?

Reviewer #2: Yes

Reviewer #3: Yes

3. Has the statistical analysis been performed appropriately and rigorously? 

Reviewer #2: Yes

Reviewer #3: Yes

4. Have the authors made all data underlying the findings in their manuscript fully available?

Reviewer #2: Yes

Reviewer #3: Yes

5. Is the manuscript presented in an intelligible fashion and written in standard English?

Reviewer #2: Yes

Reviewer #3: Yes

6. Review Comments to the Author

Reviewer #2: (No Response)

Reviewer #3: All comments have been addressed. I have no further edits or clarifications to make. Thank you very much.

7. PLOS authors have the option to publish the peer review history of their article (what does this mean?). If published, this will include your full peer review and any attached files.

Reviewer #2: **Yes: **Yaser Sarikhani

Reviewer #3: No

---

## [Editor Report · Acceptance letter]

7 Sep 2023

PONE-D-23-18471R2 

Socioeconomic status and subjective well-being: The mediating role of class identity and social activities 

Dear Dr. Jiang:

I'm pleased to inform you that your manuscript has been deemed suitable for publication in PLOS ONE. Congratulations! Your manuscript is now with our production department. 

Kind regards, 

on behalf of

Professor Petri Böckerman 

Academic Editor

PLOS ONE